# Pediatric Radial Neck Fractures: A Systematic Review Regarding the Influence of Fracture Treatment on Elbow Function

**DOI:** 10.3390/children9071049

**Published:** 2022-07-14

**Authors:** Lisette C. Langenberg, Kimberly I. M. van den Ende, Max Reijman, G. J. (Juliën) Boersen, Joost W. Colaris

**Affiliations:** 1Centre for Orthopedic Research Alkmaar (CORAL), 1815 JD Alkmaar, The Netherlands; lc.langenberg@nwz.nl; 2Department of Orthopedic Surgery, Noordwest Ziekenhuisgroep, 1815 JD Alkmaar, The Netherlands; 3Department of Orthopedic Surgery and Sports Medicine, Erasmus MC University Medical Center Rotterdam, P.O. Box 2040, 3000 CA Rotterdam, The Netherlands; kimvandenende@gmail.com (K.I.M.v.d.E.); m.reijman@erasmusmc.nl (M.R.); j.boersen@erasmusmc.nl (G.J.B.)

**Keywords:** pediatric radial neck fracture, radial neck angulation, elbow motion

## Abstract

Background: This review aims to identify what angulation may be accepted for the conservative treatment of pediatric radial neck fractures and how the range of motion (ROM) at follow-up is influenced by the type of fracture treatment. Patients and Methods: A PRISMA-guided systematic search was performed for studies that reported on fracture angulation, treatment details, and ROM on a minimum of five children with radial neck fractures that were followed for at least one year. Data on fracture classification, treatment, and ROM were analyzed. Results: In total, 52 studies (2420 children) were included. Sufficient patient data could be extracted from 26 publications (551 children), of which 352 children had at least one year of follow-up. ROM following the closed reduction (CR) of fractures with <30 degrees angulation was impaired in only one case. In fractures angulated over 60 degrees, K-wire fixation (Kw) resulted in a significantly better ROM than intramedullary fixation (CIMP; Kw 9.7% impaired vs. CIMP 32.6% impaired, *p* = 0.01). In more than 50% of cases that required open reduction (OR), a loss of motion occurred. Conclusions: CR is effective in fractures angulated up to 30 degrees. There may be an advantage of Kw compared to CIMP fixation in fractures angulated over 60 degrees. OR should only be attempted if CR and CRIF have failed.

## 1. Introduction

Although radial neck fractures in children occur frequently, there is no consensus on the optimal treatment. The indication to perform a (surgical) reduction varies widely; some authors advise striving for anatomical reduction of the radial neck, and others accept up to 60° of fracture angulation [1,2,3,4,5,6,7].

Several treatment options are available. Closed reduction (CR) without fixation, closed reduction with intramedullary pinning (CIMP) with or without pin rotation, K-wire leverage and K-wire pinning (Kw), and open reduction or combinations of the aforementioned options may be used. There is no consensus yet on the fracture angulation threshold for surgical intervention and which surgical technique should be used.

Loss of motion is reported to be the most important cause of a poor outcome [8]. Therefore, the purpose of this systematic review was to compare the elbow function following different types of treatment in relation to the angulation of the pediatric radial neck fracture. With this research, we aimed to find out which treatment modality for different types of pediatric radial neck fractures results in the best elbow function.

## 2. Materials and Methods

This study followed the guidelines of the Preferred Reporting Items for Systematic Reviews and Meta-Analyses (PRISMA). The protocol for this systematic review is registered in the PROSPERO database http://www.crd.york.ac.uk/PROSPERO/ (accessed on 7 May 2022), (registration number CRD42018088696).

### 2.1. Search Strategy

A health science librarian of our institution, with extensive experience in the conduct of literature searching for systematic reviews, assisted in designing and performing the search [9]. The following databases were searched: Embase, Cochrane Central Register, Medline, Web of Science, PubMed Publisher, and Google Scholar. The following main keywords were used: radial neck fracture, angulation, elbow, outcome, pronation, and supination. The search strategy for each database is outlined in Appendix A of this paper. The databases were searched from inception to 17 November 2021 (Figure 1).

### 2.2. Study Selection

The results from all databases were combined, and duplicate titles were removed. Two authors (K.I.M.v.d.E. and G.J.B.) screened all titles, abstracts, and full articles independently. The inclusion and exclusion criteria are listed in Table 1. Disagreements were solved by discussion, and a final decision was made by a third reviewer (J.W.C.) if there was disagreement. Patients with less than one year of follow-up were excluded.

### 2.3. Risk of Bias Assessment

The risk of bias was assessed by using the prognostic checklist adapted from the Cochrane handbook for systematic reviews, chapter 7 [10] (see Table 2). Each study was scored for selection bias, information bias, and confounding. Two authors (K.I.M.v.d.E. and G.J.B.) assessed the quality of the included studies independently. If consensus was not reached after discussion, a third reviewer (M.R.) was consulted. Finally, article quality was screened using the MINORS criteria; an overview is listed in Table 3.

Data regarding study design, number of children, age, fracture classification, type of surgical intervention or conservative therapy, range of motion (ROM) at presentation, ROM at follow-up, and complications were extracted by one reviewer (K.I.M.v.d.E.). Characteristics are listed in Table 3. The primary outcome was ROM of the elbow at follow-up.

### 2.4. Data Analysis

Each study was analyzed for individual patient data on preoperative angulation, method of treatment, and postoperative range of motion. Data regarding radial neck angulation (≤30°, 31–60°, and >60°) and treatment were extracted from articles or obtained from authors. If published articles provided insufficient patient data to be included for the data analysis, the authors were contacted with a request for individual data if contact details were available.

For each received treatment, the ROM at follow-up was evaluated (Table 4). Four different treatment groups were identified: no reduction or closed reduction only (CR); closed reduction followed by internal fixation (CRIF), either with K-wire fixation (Kw) or closed intramedullary pin fixation (CIMP); or open reduction (OR). If there was no full range of motion at follow-up (defined as at least 5 degrees of impairment in any direction described by the authors), the outcome was scored “impaired”. Differences in outcomes for several groups were statistically analyzed using the Chi-Square Fisher Exact test (*p* < 0.05) using the software SPSS Statistics for Windows, Version 27.0 (IBM Corp. Released 2020. IBM, Armonk, NY, USA).

## 3. Results

Of the 2281 publications found by our search, 52 series of pediatric radial neck fractures were of potential interest. Other than Ҫevik et al. [14], Cha et al. [15], and Cossio et al. [16], all studies showed some risk of bias. Selection bias was seen in 10 papers, information bias in 15 papers, and confounding in 18 papers (see Table 2 and Table 3).

Twenty-six case series provided sufficient data regarding angulation at trauma, treatment details, and elbow ROM at follow-up [36,37,38,39,40,41,42,43,44,45,46,47,48,49,50,51,52,53,54,55,56,57,58,59,60]. The main characteristics of the included studies are listed in Table 3. All studies had a retrospective design. In total, 551 pediatric cases could be included, ranging from 5 to 54 children per study [8,11,13,14,15,16,17,18,20,21,22,23,25,26,28,29,30,31,32,33,35]. All fractures were divided into three groups based on the degrees of fracture angulation following the classifications of O’Brien and Judet [5,7]: ≤30° (35 cases), 31–60° (247 cases), and >60° (269 cases). In total, 352 of these patients had a follow-up of at least one year. Results are depicted in Table 4, which includes data from nineteen articles [8,11,13,14,15,16,21,23,25,30,31,32].

**Table 4 children-09-01049-t004:** Data analysis of 352 pediatric patients who sustained a radial neck fracture and had at least 1 year follow-up.

Fracture Angle	N (%)	Treatment Groups	N with Loss of Motion (% of Treatment)	N (% of Angulation Group)
≤30°	25 (7.1)	CR	1 (7.7)	13 (52.0)
		CRIF CIMP Kw	0 0 0	11 (44.0) 11 (44.0) 0
		OR	1 (100)	1 (4.0)
		*Group sum*	*2 (8.0)*	
31–60°	152 (43.2)	CR	7 (63.6)	11 (7.2)
		CRIF CIMP Kw	12 (9.4) 10 (9.5) ^NS$^ 2 (9.1) ^NS$^	127 (83.6) 105 (69.1) 22 (14.5)
		OR	7 (50)	14 (9.2)
		*Group sum*	*26 (17.1)*	
>60°	175 (49.7)	CR	0	0
		CRIF CIMP Kw	33 (26.4) 30 (32.6) *^@^ 3 (9.7) *^@^	123 (70.3) 92 (52.6) 31 (17.7)
		OR	31 (59.6)	52 (29.7)
		*Group sum*	*64 (36.6)*	
>30° (31–60° and >60° combined)	327 (92.9)	CR	7 (63.6) *^	11 (3.4)
		CRIF CIMP Kw	45 (18.0) *^ 40 (20.3) ^NS^# 5 (9.4) ^NS^#	250 (76.5) 197 (60.2) 53 (16.2)
		OR	38 (55.1)	66 (20.2)
		*Group sum*	*90 (27.5)*	

CR = closed reduction without fixation or immobilization only; CRIF = closed reduction internal fixation; OR = open reduction; CIMP = retrograde intramedullar fixation; Kw = percutaneous fixation with K-wire. N = number of patients; *: significant difference; ^NS^: non-significant difference. ^$^: In the group angulated 31–60, there is no significant difference between Kw and CIMP, *p* = 0.950. ^: In the >30 angulated patients, there is a significant difference between CR (without fixation) and CRIF (CIMP and Kw) fixation; *p* < 0.001. #: In the >30 angulated patients, there is no significant difference between IM fixation or Kw fixation, *p* = 0.007. ^@^: In the >60 angulated patients, there is a significant difference between Kw and CIMP; *p* = 0.001.

Treatment options consisted of cast immobilization without reduction; closed reduction [26,27], which may be aided by leverage of a percutaneous pin [61]; K-wire fixation (either transcapitellar [18,62], across the fracture [41], by percutaneous K-wire leverage and pinning [15,16,19,29,33,46,49,52]); intramedullary K-wire [24,32]; Nancy nail or Titanium elastic nail (TEN) [11,13,22,23,25,28,34,36,54,55] combined techniques such as CIMP assisted by Kw leverage [14,17,30,43,50,53,57]; open reduction only [8]; or the description of several treatments [20,21,31,35,37,38,39,42,45,47,56,59,60]. Some included (slight) adjustments to established techniques [12,58].

Children with a fracture angulation of ≤30° who were treated with CR showed loss of motion in 7.7% at follow-up. In fractures angulated over 30 degrees, 63.6% of conservatively treated children had impaired ROM (7/11 cases). The outcome following CR was significantly worse compared to patients treated with CRIF (either Kw fixation or CIMP), in angulation >30 degrees (CR 7/11 (63.6%) impaired vs. CRIF 45/250 (18.0%) *p*-value < 0.001).

A closed reduction with intramedullary pinning was most frequently performed (250 patients in total) in both the 31–60° and >60° group. If only the groups over 30 degrees angulation are compared, there was a significantly better outcome following K-wire fixation than following CIMP (Kw vs. CIMP 5/53 (9%) vs. 40/197 (20%); *p*-value <0.001). This is also true for a separate analysis of the >60° group (K-wire vs. CIMP 3/31 (9.7%) vs. 30/92 (32.6%) impaired, *p*-value of 0.001), but a separate analysis of Kw vs. CIMP in the 31–60 group yields a non-significant difference (Table 4). Overall, there was no significant difference between Kw and CIMP.

Open reduction resulted in an impaired range of motion in about 60% of cases. All but one OR had been performed in fractures angulated over 30°. Nine separate articles published data on open reductions (OR), but the numbers were too small for a statistical analysis. Following OR, there had been 7 fractures without fixation, 11 with IM fixation, and 18 with K-wire fixation; 3 were not described in detail.

## 4. Discussion

To our knowledge, this is the first systematic review that performed a pooled analysis that focused on range of motion as an outcome following pediatric radial neck fracture treatment. Overall pediatric radial neck fractures resulted in impaired elbow function in 26% of cases. Radial neck fractures with an angulation of ≤30° demonstrated good results with CR. Fractures angulated >60° showed the least ROM impairment if K-wires were used. Open reduction had been mostly used in severely angulated fractures and often ended in an impaired elbow function.

In the literature, there is a wide variety of different scales and ratings to report radial neck fracture outcomes [63]. Only a few authors used a validated outcome scale, such as the Mayo Elbow Performance Scale (MEPS) or the (quick)DASH (Table 3). Many used their own rating system to judge the clinical or radiological outcome, which led to low comparability. Thereby, most authors only published the mean outcomes of certain groups of patients or mixed outcomes of several fracture classifications. Data pooling of individual patient cases and a meta-analysis for various treatments and their outcomes were hence impossible; the only analysis that could be performed on the extracted data was an evaluation of the outcome of range of motion based on fracture classification. 

### 4.1. CR and Indication for Fracture Reduction

All children who were treated with immobilization or closed reduction only received a long arm cast (or collar and cuff sling under the clothes [21]). In the studies of Fowles and Kassab [18] and Jones and Esah [21], the elbows were immobilized for 3 weeks. For all other conservative treatments, the duration of immobilization was unclear [20,26,31].

Overall, no manipulation was performed when the initial angle was ≤30°, and closed reduction was indicated when the initial angle was >30°. An exception is an article by Jones [21] that advised fracture reduction when angulation exceeded 15 degrees. Closed reduction was always unsuccessful in radial neck fractures angulated over 60°, following a study that evaluated the success rate of closed radial neck fracture reduction in the emergency ward [64]. The same article stated that delayed reduction that was attempted over 24 h following trauma, may be prone to failure.

Closed reduction of a radial neck fracture may be challenging and might result in residual angulation or re-displacement. In a series of 48 fractures that were reduced by closed manipulation without fixation, as many as 36 fractures remained slightly or severely unreduced [45]. The quality of the cast may play a role, which may be calculated using the casting index (CI) [65,66].

Closed reduction without osteosynthesis in fractures angulated >30 degrees showed loss of motion at follow-up in over 60% of cases (N = 11). A recent review therefore advised to consider the percutaneous fixation of a successful closed reduction [63]. Malunion was the main reason for loss of motion in this group [42,43,51]. Some stated that closed manipulation may be attempted in fractures angulated as much as 45 degrees, but if residual angulation exceeds 20 degrees, intramedullary pin fixation should be considered [25]. Others concluded that closed reduction should be considered unsuccessful if residual angulation is over 15 degrees [21].

Closed reduction may be aided by percutaneous K-wire manipulation [28,29,52,67]. Some authors, however, stated that the manipulation by the K-wire in proximity to the physis may cause abnormalities to the physis or risk of neurological damage, and advised against it [32]. Small series were published which demonstrated other options to facilitate reduction: a forceps may be introduced ulnar to the radial neck [58], or a small elevator may be introduced at the fracture site [19]. Nevertheless, in the series that described an elevator-assisted reduction technique, a premature physis fusion occurred in 4/10 patients.

### 4.2. Choice of Treatment and Relation to Postoperative ROM

Although one of the case series showed favorable results of CIMP [30], our combined analysis shows that there is a better ROM following K-wire fixation compared to CIMP fixation, which is significant in fractures angulated >60 degrees. Potentially, this difference may be explained by the low number of patients reported in single case series, which renders a high risk of bias.

### 4.3. Open Reduction (OR)

OR should only be performed when closed reduction fails. For example, the introduction of an intramedullary device may be challenging if angulation exceeds 80° [68]. A small incision (<3 cm) is recommended [20], the annular ligament should be preserved, and instruments that could damage the radial head during reduction should be avoided. The use of a “Joy stick” K-wire in the proximal fragment to aid fracture reduction is favored over the use of clamps to prevent potential damage to the posterior interosseous nerve (PIN) [69].

Poor results following OR may be caused by damage of the blood supply [70], proximal radioulnar joint adhesion [8], or periarticular ossification [57,71]. Potentially, the focal damage to tissues due to trauma plays a role [8]. Nevertheless, interposing soft tissue makes open reduction necessary in some cases [32]. In all fracture angulation groups, loss of motion was seen in about half of the children treated with open reduction. We therefore agree with Klitscher et al. [22], who stated: “Every manipulative technique should be tried before open reduction is chosen”. However, there may be bias because OR was sometimes described as the last available option when closed reduction failed.

Although some authors stated that if the radial head was stable following OR, fixation was not always necessary [8], this is disputed by our high percentage of loss of ROM in non-fixated fractures after OR. Given the fact that this percentage is significantly lower in the CRIF groups, fixation by an intramedullary nail or K-wire fixation should be considered, even after OR.

The results of this systematic review are subject to some limitations. First of all, the overall level of evidence is low. Almost all articles were level 4, and some were level 3. The overall quality of included articles is mediocre, with a risk of selection bias in 10 articles, information bias in 15 articles, and confounding bias in 18 articles. Several articles described a new or modified surgical technique without a power analysis for group size [32,33], and without a statistical analysis to compare to traditional techniques or a clear comparative design. The low incidence of displaced radial neck fractures and subsequently small cohort sizes played a role. The scores for the MINORS criteria were low for all studies, mainly due to the retrospective character of all case series that were included.

Secondly, this article only focuses on fracture angulation without considering the effect of fracture translation or rotation. In addition, associated injuries, such as ipsilateral olecranon fracture, ipsilateral fracture of the medial epicondyle, or elbow instability, were not registered; however, they can be present in 50% of children suffering a radial neck fracture [31,39]. The influence of the presence of a more extensive injury on the choice of treatment for the radial neck fracture [17] or the postoperative outcome [25,40] is still subject to discussion [18]. Some authors stated ROM would not be impaired [42]; others disagreed and showed a less favorable prognosis when associated injuries were present [17,39,42,48].

Thirdly, growth can behave like a friend or an enemy in children and might affect the outcomes. Nevertheless, a radial neck fracture is near the minimal active proximal physis, which results in less correction than in distal radius fractures. To minimize the influence of correction by growth, we only included children with a minimal follow-up of one year.

The influence of immobilization in the non-conservative groups could not be calculated. There was no evidence that postoperative immobilization had any advantages [57,72], and the worse outcome in ROM was seen when the elbow was immobilized for more than three weeks [37]. It is noteworthy that almost none of the included studies mentioned physiotherapy. Only Wang et al. [57] described “exercises under supervision (…) 1 day after operation”, and Walcher et al. [33] mentioned physiotherapy only in one complex case.

## 5. Conclusions

This systematic review shows that conservative treatment with or without the closed reduction of pediatric radial neck fractures with primary angulation up to 30° results in good elbow function. In radial neck fractures with an angulation of >60°, closed reduction followed by K-wire fixation may have an advantage over intramedullary fracture fixation, but this difference is not significant in fractures angulated 31–60°. Open reduction should only be performed if closed reduction fails, and caution should be taken not to (further) damage the physis and radial head vascularization.

## Figures and Tables

**Figure 1 children-09-01049-f001:**
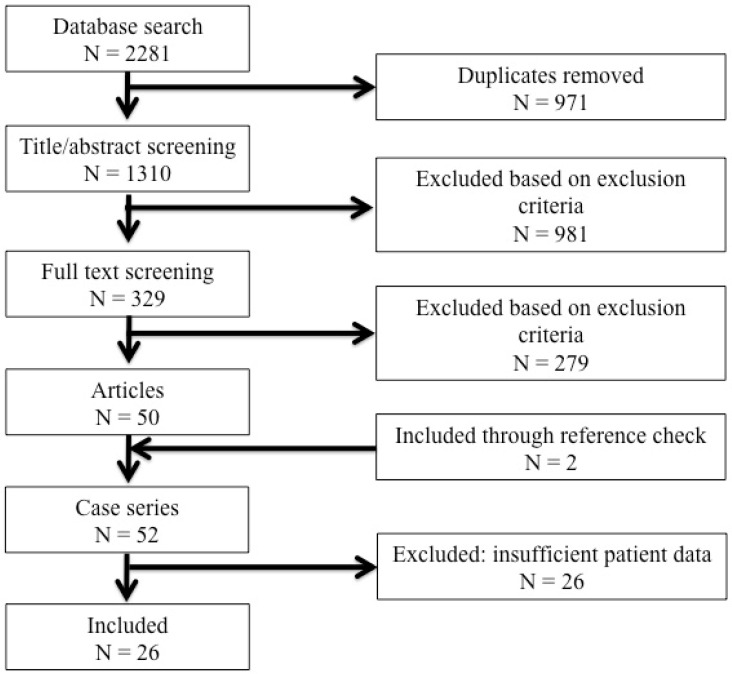
PRISMA-guided systematic search.

**Table 1 children-09-01049-t001:** Inclusion and exclusion criteria.

Inclusion Criteria	Exclusion Criteria
Prospective or retrospective follow-up study	Review/meta-analysis
≥5 children with radial neck fractures	Age > 16 years
Fracture angulation should be reported	Elbow prosthesis
Radiological imaging at presentation	Animals
Outcome: range of motion at follow-up	Less than one year follow-up
Outcome linked to fracture angulation and treatment	
Language: English, Dutch	

**Table 2 children-09-01049-t002:** Risk of bias assessment based on adapted Cochrane checklist.

Author	Selectionbias	Informationbias	Confounding
Al-Aubaidi (2012) [11]			
Bilal (2021) [12]			
Brandão (2010) [13]			
Ҫevik (2018) [14]			
Cha (2012) [15]			
Cossio (2014) [16]			
Endele (2010) [17]			
Falciglia (2014) [8]			
Fowles (1986) [18]			
Futami (1995) [19]			
Gutierrez-de la Iglesia (2015) [20]			
Jones (1971) [21]			
Klitscher (2009) [22]			
Koca (2017) [23]			
Massetti (2020) [24]			
Metaizeau (1993) [25]			
Monson (2009) [26]			
Shah (2020) [27]			
Stiefel (2001) [28]			
Tanagho (2015) [29]			
Tarallo (2013) [30]			
Tibone (1981) [31]			
Ugutmen (2010) [32]			
Walcher (2000) [33]			
Yallapragada (2020) [34]			
Zhang (2016) [35]			

Scores: green = low risk, orange = moderate risk, red = high risk.

**Table 3 children-09-01049-t003:** Part a: Overview of included articles, part b: MINORS criteria.

(a)												
Author	Year		Fracture Classification	Injury Type			N	1y FU	Mean Age in Years (Range)		Mean Follow-Up in Years (Range)	Outcome
Al-Aubaidi [11]	2012		Steele	all pt w/open physis treated with metaizeau			16	16	12 (9–15)		3.3 (1.3–6.3)	DASH
Bilal [12]	2021		>30°	>30°,intramedullary nailing (TEN)			15	15	10.1 (6.4–15.8)		2.1 (1.3–3)	Tan&Mahadev
Brandão [13]	2010		O’Brien	O’Brien type 3			28	26	8.6 (6–13)		4.3 (1.7–10)	radiologic union, ROM
Ҫevik [14]	2018		Judet	Judet 3/4			20	20	9.75 (4–13)		2.9 (1.1–7)	ROM
Cha [15]	2012		Judet	Judet 3/4			13	13	10.4 (6–13)		3.5 (2.4–4.4)	flynn score
Cossio [16]	2014		Judet	Judet 3/4			9	9	9.1 (6–12)		2.2 (1–3)	Tibone
Endele [17]	2010		Judet	all RN# in a retrospective period			54	42	8 (1-13)		4 (0.5–11)	ROM
Falciglia [8]	2014		O’Brien	all RN# in a retrospective period without success of CR or KW			24	24	7.1 (4.3–10.2)		7.1 (3.2–12.1)	MEPS
Fowles [18]	1986		<20°, >20°	all RN# in a retrospective period			23	17	9.1 (5–13)		1.5 (0.7–2.8)	ROM
Futami [19]	1995		none	angulated RN# (not specified)			10	10	9 (6–13)		u	Tibone and Stolz
Gutierrez-de la Iglesia [20]	2015		Judet	Judet 3/4			51	0	8 (3–15)		1.2 (0.7–3.3)	Tibone and Stolz, Ursei
Jones [21]	1971		15–29°, 30–59°, 60–90°	all RN# in a retrospective period			34	18	10 (5–13)		5 (1–14)	Steele
Klitscher [22]	2009		Judet	Judet 3/4			28	0	8 (5–11)		2.7 (0.5–5.6)	MEPS, Metaizeau
Koca [23]	2017		Judet	Judet 3			11	11	7.7 (6–10)		2.0 (1.7–2.7)	Leung/Peterson
Massetti [24]	2020		judet	Judet 3/4			20	0	7.8 (2–11)		0.7–3.8	MEPS
Metaizeau [25]	1993		Judet	Judet 3/4			47	47	10.7 (5–13)		4 (ns)	MEPS
Monson [26]	2009		Degrees	all RN# in a retrospecitve period			6	6	9.5 (6–11)		0.36	Morrey, Metaizeau
Shah [27]	2020		Judet	Judet 4			10	10	8.6 (6–12)		1 (0.8–1.3)	Steinberg, Rodriguez-Merchan
Stiefel [28]	2001		Judet	Judet 4			6	6	8.4 (7–10.8)		u (0.75–2.5)	ROM
Tanagho [29]	2015		Steele	isolated metaphyseal RN# >30°			9	9	9.6 (u)		1.6 (u)	Own rating system
Tarallo [30]	2013		Judet	Judet 3/4			20	20	11 (6–16)		3.5 (1.3–5.3)	MEPS, Metaizeau
Tibone [31]	1981		Degrees	all RN# in a retrospective period			23	23	9.2 (4–14)		3.15 (2.0–8.0)	ROM
Ugutmen [32]	2010		Judet	RN# with open growth plates			16	16	8 (6–13)		2 (1.5–3.3)	Metaizeau
Walcher [33]	2000		Judet	Judet 2/3, failed CR			5	0	7 (u)		3 (u)	ROM, own rating system
Yallapragada [34]	2020		Judet	Judet 3/4			21	0	8 (u)		0.4 (0.3–0.5)	OES, Metaizeau
Zhang [35]	2016		Judet	Judet 3/4			50	0	8.4 (5.6–13)		2 (u)	MEPS
							569	352	8.96		2.69	
(b)												
Author		Year		MINORS Total	Aim	Consecutive Cases		End Points	Bias	Follow-Up	Lost to FU	Study Size
Al-Aubaidi [11]		2012		7	1	2		1	0	2	1	0
Bilal [12]		2021		6	1	1		2	1	2	0	0
Brandão [13]		2010		10	2	2		2	1	2	1	0
Ҫevik [14]		2018		10	2	2		2	2	2	0	0
Cha [15]		2012		10	2	2		2	2	2	0	0
Cossio [16]		2014		8	1	1		2	2	2	0	0
Endele [17]		2010		9	1	1		2	1	2	2	0
Falciglia [8]		2014		8	2	1		2	1	1	1	0
Fowles [18]		1986		8	1	2		1	0	2	2	0
Futami [19]		1995		2	0	0		0	0	2	0	0
Gutierrez-de la Iglesia [20]		2015		8	2	1		2	1	2	0	0
Jones [21]		1971		8	1	1		1	1	2	2	0
Klitscher [22]		2009		9	1	2		2	1	2	1	0
Koca [23]		2017		8	2	1		2	1	2	0	0
Massetti [24]		2020		5	2	1		1	2	1	0	0
Metaizeau [25]		1993		7	2	1		2	1	1	0	0
Monson [26]		2009		3	1	1		1	0	0	0	0
Shah [27]		2020		4	0	2		1	1	1	0	0
Stiefel [28]		2001		4	2	1		1	0	0	0	0
Tanagho [29]		2015		4	1	1		0	0	1	1	0
Tarallo [30]		2013		8	2	1		2	1	2	0	0
Tibone [31]		1981		9	1	2		2	1	2	1	0
Ugutmen [32]		2010		5	1	1		1	1	1	0	0
Walcher [33]		2000		4	1	1		0	1	1	0	0
Yallapragada [34]		2020		4	1	1		1	1	1	0	0
Zhang [35]		2016		6	2	1		1	0	2	0	0

u = unknown; RN# = radial neck fractures; ROM = range of motion; MEPS = Mayo Elbow Performance Score; OES = Oxford Elbow Score.

## Data Availability

Access to the database may be requested by email via the corresponding author.

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
