# Peer review of "Pediatric Radial Neck Fractures: A Systematic Review Regarding the Influence of Fracture Treatment on Elbow Function"

_children, 2022, doi:10.3390/children9071049_

Round 1

Reviewer 1 Report

Dear Authors the topic is estremely interesting.

As regards the introduction from line 27 to line 35, i suggest to improve the concept linked to surgical indication in pediatric patients. In fact according to bibliography, Authors analyzed   the relationship between conservative and surgical treatments in terms of functional results according age stratification. For this reason i suggest to cite the following article in which the Authors analyzed the results of treatment of forearm pediatric fractures.

“Failure Predictor Factors of Conservative Treatment in Pediatric Forearm Fractures

Biomed Res Int. 2018 Jul 12;2018:5930106. doi: 10.1155/2018/5930106. 

Maccagnano G. et al”

As regards M&M the strategy  and methods (prisma guidelines) are well described.

As regards the discussion is supported by results.

As regards the conclusion the Authors underlined the importance of close reduction. The open reduction and k-wire fixation for pediatric patient have to consider only in case of  failure of conservative treatment

Author Response

Dear reviewer, 

Thank you for your time and effort in reviewing our manuscript. 

Guided by your suggestions, the literature analysis regarding choices in conservative treatment has been expanded. The article by Maccagnano et al has been added as a referenc and a sentence regarding the potential influence of cast quality has been added. 

Reviewer 2 Report

I want to compliment with the Authors for the outstanding work in their systematic review. The paper is well written and the limitations are explained very well.

Author Response

Dear reviewer,

Thank you very much for your time and effort invested in reviewing our manuscript. 

Reviewer 3 Report

You wrote that patients with FU of less than 1 year were excluded, but this is not mentioned in table 1 as exclusion criteria. These articles were excluded at the very end. Did you consider excluding them at step no 3?

Author Response

Dear reviewer,

Thank you for your time and effort in reviewing our manuscript. 

In a late phase of writing, we decided to exclude the patients with less than one year follow-up, to increase the quality of the database. Thank you for your suggestion to adjust the exclusion criteria accordingly, table 1 has been corrected.

Yours sincerely, 

Lisette Langenberg